# What are the barriers to the diagnosis and management of chronic respiratory disease in sub-Saharan Africa? A qualitative study with healthcare workers, national and regional policy stakeholders in five countries

Stephen Mulupi ,[1,2] Irene Ayakaka,[1,3] Rachel Tolhurst,[1] Nicole Kozak,[1,4] Elizabeth Henry Shayo,[1,5] Elhafiz Abdalla,[6] Rashid Osman,[6] Uzochukwu Egere,[1] Stellah G Mpagama ,[7] Martha Chinouya,[8] Kingsley Rex Chikaphupha,[4] Asma ElSony,[6] Helen Meme,[2] Rose Oronje,[9] Nyanda Elias Ntinginya,[10] Angela Obasi ,[1,11] Miriam Taegtmeyer[1,12] On behalf of the NIHR International Multidisciplinary Programme to Address Lung Health and TB in Africa (IMPALA) Consortium

For numbered affiliations see end of article.

**Correspondence to**
Stephen Mulupi;
Stephen.Mulupi@lstmed.ac.uk

## ABSTRACT

**Objectives** Chronic respiratory diseases (CRD) are among the top four non-communicable diseases globally. They are associated with poor health and approximately 4 million deaths every year. The rising burden of CRD in low/middle-income countries will strain already weak health systems. This study aimed to explore the perspectives of healthcare workers and other health policy stakeholders on the barriers to effective diagnosis and management of CRD in Kenya, Malawi, Sudan, Tanzania and Uganda.

**Study design** Qualitative descriptive study.

**Settings** Primary, secondary and tertiary health facilities, government agencies and civil society organisations in five sub-Saharan African countries.

**Participants** We purposively selected 60 national and district-level policy stakeholders, and 49 healthcare workers, based on their roles in policy decision-making or health provision, and conducted key informant interviews and in-depth interviews, respectively, between 2018 and 2019. Data were analysed through framework approach.

**Results** We identified intersecting vicious cycles of neglect of CRD at strategic policy and healthcare facility levels. Lack of reliable data on burden of disease, due to weak information systems and diagnostic capacity, negatively affected inclusion in policy; this, in turn, was reflected by low budgetary allocations for diagnostic equipment, training and medicines. At the healthcare facility level, inadequate budgetary allocations constrained diagnostic capacity, quality of service delivery and collection of appropriate data, compounding the lack of routine data on burden of disease.

**Conclusion** Health systems in the five countries are ill-equipped to respond to CRD, an issue that has been brought into sharp focus as countries plan for post-COVID-19 lung diseases. CRD are underdiagnosed, under-reported and underfunded, leading to a vicious cycle of invisibility and neglect. Appropriate diagnosis and management require health systems strengthening, particularly at the primary healthcare level.

## STRENGTHS AND LIMITATIONS OF THIS STUDY

⇒ The five studies were conducted independently by the country study teams, allowing them to take local health system factors in each country into account when interviewing.

⇒ The inclusion of five countries with very similar emerging themes allows for comparison between countries and gives a sense of how widespread barriers to chronic respiratory disease (CRD) are.

⇒ We were able to triangulate perspectives of different types of study participants, enhancing rigour in our analysis.

⇒ Differences in sampling the study sites, for example, number and levels of hospitals and rural/urban variations limited comparisons.

⇒ We were not able to get patients' perspectives on demanding and accessing health services for CRDs.

## INTRODUCTION

Chronic respiratory diseases (CRD) account for approximately 4 million deaths per year worldwide.[1 2] Globally, the most prevalent CRDs are asthma and chronic obstructive pulmonary disease (COPD)[3] but the definition includes other non-infectious lung and airways diseases such as bronchiectasis and post-tuberculosis (TB) lung disease. The

burden of CRD morbidity and mortality is rising steeply in low/middle-income countries (LMICs).[4 5] The high prevalence of CRD in sub-Saharan Africa (SSA) is driven in part by high rates of recurrent childhood respiratory infections and pulmonary TB, which are important precursors of CRD.[6–9] A hospital-based study in Tanzania among post-TB patients estimated a 74% prevalence of abnormal lung function.[10] Other common risk factors include exposure to tobacco smoke, indoor pollution from biomass fuels and occupational exposures in the mining industry, sugar and tobacco plantations.[11 12] In Uganda, the age-adjusted prevalence for any chronic respiratory condition in both rural and urban settings is 20%,[13] and in Malawi over 40% of adults randomly sampled in an urban population had abnormal lung function.[14]

The management of CRD in SSA is undermined by weak health systems characterised by lack of appropriate medical equipment and low diagnostic capacity.[15 16] Healthcare workers trained in the management of CRD are scarce and medicines are either unavailable or unaffordable by many patients.[17 18] Existing lung healthcare pathways in SSA focus on TB with smear-negative patients repeatedly visiting health facilities or being referred from one facility to another seeking definitive diagnosis. A wide range of policy design, implementation and service delivery gaps undermine CRD care pathways. Importantly, there is limited understanding in how specific policy and health facility-level factors interact and shape health service delivery in SSA contexts.

In this qualitative study, we sought to understand and characterise the views and perspectives of policymakers and healthcare workers on CRD policies and services in five countries in SSA. We present findings that show clear linkage of policy-level factors to service delivery experiences. We contribute evidence to inform the design of future policy interventions to strengthen the delivery of services and ultimately improve CRD management in similar settings. The five linked studies formed baseline assessments for country-level research within the National Institute for Health Research, International Multidisciplinary Programme to Address Lung Health and TB in Africa (IMPALA; https://www.lstmed.ac.uk/impala).

## METHODS

Five studies were independently conducted in Kenya, Malawi, Sudan, Tanzania and Uganda between 2018 and 2019 based on a constructivist philosophical paradigm using qualitative study methods[19 20] to collect data and to perform data validation, analysis and interpretation to answer the study questions. Ministry of Health (MoH) officials and other healthcare policy stakeholders were purposively sampled for their roles and experiences in policy design and implementation alongside health workers' experience in the management of CRD (table 1). To secure interview appointments, participants were contacted through email, phone or verbally, as appropriate. Public healthcare facilities were selected to include both urban and rural settings and different facility levels (table 2). All served populations with high prevalence rates of TB and other lung conditions.

### Data collection

Key informant interviews (KIIs) with policy stakeholders (online supplemental appendices 1–3) explored opinions about prioritisation of CRD; (un)availability and operationalisation of CRD policies; systemic factors enhancing or impeding provision of healthcare services for CRDs; and opinions on how systems could be improved. In-depth interviews (IDIs) with healthcare workers (online supplemental appendices 4–6) explored their experiences in diagnosing and managing CRD; availability of diagnostic equipment and medicines; experience of training; and their perceptions of what has worked well or not, in management of CRD. Semi-structured interview guides were developed broadly aligning with the health systems' building blocks.[21] Interviews were conducted from May 2018 to March 2019, by experienced qualitative researchers in English, or the local languages as appropriate, and digitally recorded with consent. These face-to-face interviews took place in quiet, private rooms within the workplaces of the participants, to protect participants' confidentiality, minimise disruptions and ensure quality of audio recordings. The discussions took around 40 minutes. The researchers in Sudan, Malawi and Tanzania had no prior relationships with the participants. In Malawi, a female white Canadian researcher (NK) conducted some of the interviews (others were done by a Chichewa-speaking research assistant) and came from an outsider perspective. The researcher from Uganda (IA) is a respected female medical doctor who was known to some of the participants. This positionality required critical reflection on being an 'insider' to the health system during the research period. Similarly, in Kenya, affiliation of the male researcher (SM) to the Kenya Medical Research Institute, a government agency,

**Table 1** The distribution of participants across the five case study countries

| Informants | Kenya | Malawi | Sudan | Tanzania | Uganda |
|---|---|---|---|---|---|
| National, regional and district-level policy stakeholders | 15 | 13 | 14 | 13 | 5 |
| Healthcare workers | 14 | 5* | 14 | 10 | 6 |
| Total | 29 | 18 | 28 | 23 | 11 |

*Senior clinicians.

**Table 2** Distribution of healthcare facilities in the five case study countries

| Public health system level | Kenya | Malawi | Sudan | Tanzania | Uganda | Total |
|---|---|---|---|---|---|---|
| Primary healthcare facilities (health centre and dispensary level) | 3 | 1 | 2 | 8 | 6 | 20 |
| District-level hospitals | 1 | 1 | 8 | 1 | 0 | 11 |
| National/regional referral hospitals | 0 | 2 | 0 | 1 | 2 | 5 |
| Total number of facilities | 4 | 4 | 10 | 10 | 8 | 36 |

may have shaped trust and response to the study, though the effect on issues is unknown. The researchers jotted notes about participants' comments and the researcher's own thoughts during the interviews; they used memoing as soon as possible after an interview and during the transcription of recordings. Recordings were transcribed verbatim and translated to English for analysis. Field notes complemented the audio recordings.

### Intercountry analysis

Data collection was preceded by a common training in policy (2018) for the researchers and followed by joint analysis meetings that discussed similarities and differences between country-specific study findings; the codes and topics were identified within each country and combined during analysis and the respective country teams reread transcripts to confirm accuracy and identify emerging themes. Authors SM, IA, RT, MT and EHS participated in the coding meetings and determination of the final codebook. Initial codebooks were developed from the broad interview guide topics, then updated inductively as novel codes emerged. A final codebook was discussed, further refined and applied to the data, using the framework approach,[22] with analysis supported by NVivo V.11.0 (QSR International 1999) software. The main themes and subthemes were organised through NVivo nodes and subnodes, respectively, and subsequently into charts. An exploration of emerging patterns led to the identification of final themes with selected quotes used to illustrate specific findings.

### Findings

The WHO, in 2007, proposed a framework describing health system functions in terms of six core interdependent components or 'building blocks': (1) service delivery; (2) health workforce; (3) health information systems; (4) access to essential medicines; (5) financing and (6) leadership/governance.[21] Effective coordination and optimal performance of these health system blocks would theoretically enhance achievement of health system goals, that is, 'improving health and health equity, in ways that are responsive, financially fair, and make the best, or most efficient, use of available resources'.[21] We used the WHO health systems building blocks to frame our exploration at strategic and service delivery levels. A picture emerged of neglect of CRD at each level of the systems and for each building block with consequences across the whole system.

### Strategic level

#### Variable availability and awareness of policy

CRD policy availability varied between countries. Kenya and Tanzania have fully developed and adopted policy strategies for lung health[23 24] and the national and district stakeholders were aware of this; however, few healthcare workers were aware of the policies or where to find them. There was no standalone policy for CRD in Uganda and in Malawi; CRD was embedded within non-communicable disease (NCD) policy, which was in draft, with limited awareness among all levels of participants. In Sudan, there was no specific policy for CRD.

> I do not know whether there is an existing policy for TB and CRD. There is a general strategy; the national health policy that generally addresses chronic diseases. (KII-F, Sudan Federal-MoH official)

#### Lack of CRD data

In all five contexts, policy stakeholders associated the limited data on burden of disease with low visibility and low domestic budget allocations for CRD as shown by this typical quote:

> For COPD we don't have data, we don't know its prevalence, we don't know its incidence, we don't know its mortality. (IDI national informant, Tanzania)

None of the study countries had comprehensive population surveys on CRD, but Sudan, Kenya and Uganda reported research evidence about asthma, COPD and the growing burden of CRD. In Malawi, participants noted that CRD were not included in the recent nationwide burden of disease study and went further to highlight the lack of data on cost-effectiveness.

> So [CRD] never feature in the cost effectiveness analyses that allow for their prioritization, benchmarked against the other priorities where there is evidence….for chronic respiratory disease we have to recognize that the evidence base for the country is very tiny. So being able to advocate beyond, the NCD Department into the Treasury, into the Health Sector Strategic Plan, is multiple steps away. (Malawi-KI 18, researcher)

Neither were routine data able to fill the evidence gaps due to a lack of appropriate data collection tools and shortages of health information and records officers. Participants in all the five countries emphasised the need

to address these gaps in developing investment cases for CRD at national level.

> You perform a study and show them the results. How many are affected? How many are disabled? How many lose their jobs? How many houses lose support? If you provide such work, you might be able to convince the (national treasury) officials. (KII-S5, Gezira State, Sudan MoH official)

### Lack of donor prioritisation decreases budgetary allocation
The health system financing arrangements in all the study countries were reported as being heavily dependent on external partners. CRD were perceived to be given low priority by donors because of their generally non-infectious nature and the perception that they were less fatal than infectious diseases such as malaria, HIV/AIDS and TB, and maternal and child health conditions. Even within the NCD departments in the study countries, efforts and resources are majorly directed towards cancer, diabetes, hypertension and other cardiovascular diseases.

> We have some reasonable data on the burden of asthma, but asthma doesn't do one thing, doesn't kill people that much and so because it doesn't kill, it doesn't get the attention it should get… everybody gets worried about things that kill (TB, malaria, HIV/AIDS, diabetes, cancer) and not those that give you chronic illness. (Kenya, KII-KD)

### Current priorities
Positive steps in data collection and policy development were also described. In response to a global spotlight on NCDs and recent increased donor interest, the Kenyan and Tanzanian governments plan to close the evidence gap by capturing CRD data in routine health facility tools, and subsequent nationwide surveys like the Demographic and Health Surveys.

> We are refining the tools to better capture the NCDs and have specific NCD registers like in the general outpatient settings. (KII, Kenya-MoH, national level)

At the time of data collection, Kenya was in the process of ratifying the protocol to eliminate illicit trade in tobacco products, and Uganda had banned tobacco smoking in public, tobacco advertising, promotion and sponsorship, and limited sales. This suggests a heightened prioritisation of an important risk factor of CRD on the policy agenda, with potential for downstreaming policy interventions.

### Service level
Key service-level barriers arising from policy failures included challenges in accessing CRD diagnostics and medicines, limited reporting, and low confidence and skill in diagnosis and management. There was a disconnect between the perceptions of policy stakeholders, which reflected what was available on paper, and the front-line health workers, who reported feeling ill-equipped to attend patients in practice.

### Lack of diagnostic equipment
Participants universally agreed that the entry point to CRD diagnosis in their TB-endemic contexts required the exclusion of active TB disease in chronic cough. Diagnostic capacity was defined as a combination of sputum screening, with the availability of radiology, lung function testing, skilled staff and diagnostic and treatment algorithms for the common CRD (asthma, post-TB lung disease and COPD). While each study country had chest clinics at tertiary hospitals with some level of equipment and expertise, only TB screening had been systematically decentralised at lower levels.

Challenges were described in each aspect of diagnosis, with patients often paying out-of-pocket costs. All countries reported delays in sputum screening. In some instances, these delays extended to more than a week and were associated with additional costs to patients and subsequent loss to follow-up.

Finally, the widespread lack of (or dysfunctional) X-ray equipment and lack of qualified personnel reported in lower-level public facilities in Kenya, Malawi, Uganda and Tanzania meant that patients had to seek chest X-rays (and CT scans) from private healthcare providers, whose costs were unaffordable to most. The lack of peak flow metres and spirometry machines meant that patients requiring lung function testing were referred to the national teaching and referral hospitals.

> Yeah, there was a time, we also [provided spirometry] but …our machine broke down … when we see they are needing that spirometry we send them to Kenyatta National Hospital. (Healthcare worker, Kenya)

In Tanzania, participants noted that despite the recommendation to have peak flow metres at the primary healthcare level, none were available and not a single facility (including the referral hospital) offered spirometry. In Kenya, Malawi, Sudan and Uganda, the use of spirometry was reported to be limited to tertiary hospitals with international research collaborations, where training and equipment are maintained through grants from external partners. Although some policy-level stakeholders thought otherwise, health workers at the lower-level hospitals in all countries expressed dissatisfaction with the lack of peak flow metres and spirometers. In Malawi, there were conflicting reports on the distribution of spirometry services across facilities. While policymakers claimed that all tertiary hospitals had spirometers, healthcare workers reported that spirometers were only available at one teaching hospital as part of a research project.

### Lack of CRD training and guideline dissemination
According to majority of key informants in Kenya, Uganda and Tanzania, most healthcare workers had not

received training on the latest diagnostic procedures for CRD, including the use of spirometry:

> Honestly, we have not received any training for CRD; we used to go for TB trainings only. (IDI, healthcare worker, Tanzania)

Malawian participants described a successful pilot decentralising CRD training, but this was yet to be scaled up nationally. Participants in all contexts emphasised the need to improve in-service training countrywide. In-service training opportunities were ad hoc and often 'erratic' (Sudan), mostly supported by specific programmes, including research projects (eg, Uganda), non-governmental organisations and pharmaceutical companies promoting specific products, and characterised by low coverage of necessary staff.

> …we started a program with some collaborative partners, … they bring health workers from some districts, and we come and do modules to understand for example what asthma is, how its diagnosed what is the best treatment, because in [medical] school frankly, people don't learn so much for the benefit of the wider community. (MoH consultant physician, Uganda)

Most of the available training focused on TB. In Tanzania, this was to the exclusion of CRD. In Kenya, CRD was added on to TB training. Healthcare workers in Kenya reported difficulties in accessing professional training, within the devolved government set-up. Lack of financial sponsorship from government, and a requirement that they resign if they applied for full-time study, imposed steep opportunity costs.

Standard processes of communicating guidelines from national policy level to service delivery levels were felt to be inadequate; for example, healthcare workers in Kenya described learning about asthma management guidelines incidentally, during internet use or informal discussions with colleagues. There was limited awareness of treatment guideline updates with many clinicians still prescribing available oral salbutamol instead of inhalers, despite lack of evidence for efficacy. Likewise, in Tanzania, although the CRD guidelines were available on the MoH website and copies were seen at the sites, participants seemed unaware of their content.

### Limited availability of drugs and lack of confidence in management

Restrictive policy guidelines and user fees imposed barriers to accessing corticosteroid inhalers. For example, beclomethasone is classified as a central facility-level medicine in Malawi that required international procurement protocols prone to delays of 4–7 months. In Tanzania, only bronchodilators, injectable and oral steroids for acute asthma are allowed at lower levels of service provision. Similarly, in Uganda, some medicines were restricted at the primary level. In contrast, in Kenya, bronchodilators and corticosteroids have been included in pharmacy order forms for primary healthcare facilities where they are provided free, implying shifts in practice to enhance access. In Sudan, asthma medications including inhalers and corticosteroids are supposed to be provided free of charge especially for patients in acute attacks. However, these are often not available in emergency rooms at both district and national levels, and when available, they are not always free of charge.

> Some of the services for asthma might be provided for free in the emergency room, but the majority of it is not provided for free although it should [be]. (KII-S4, Gezira State MoH official)

Additionally, respondents from all five countries reported frequent drug stockouts even of the few CRD treatments included on essential medicines lists. At the time of data collection, the Kenya Medical Supplies Authority had stopped supplies to counties with outstanding bills. At primary care level in Tanzania and Uganda, drugs other than oral salbutamol were unavailable.

Front-line clinicians at primary and district level felt ill-equipped to meet the complexity of cases. Healthcare workers explained that they lacked confidence in managing chronic CRD, although acute asthma attacks would be treated (nebulisers, oral aminophylline and epinephrine injections were all mentioned).

> We end up using treatments we are not supposed to (for example oral salbutamol) because that is what is available. (Uganda, clinical officer HC III)

Key informants in Kenya and Uganda mentioned measures to enhance treatment capacity through review of the curricula for in-service training of clinical and medical officers, and for Kenya, the development of 'module 13' for community health volunteers, on identifying and managing asthma.

> We are working closely with KMTC [Kenya Medical Training College] itself, the Kenya Clinical Officers Council and the Nursing Council of Kenya to give NCD's more prominence in the pre-service training. (KII-MoH, Kenya)

Most healthcare workers, however, referred chronic asthma and other undiagnosed CRD conditions to higher-level facilities, but even at this level, limited capacity for management was described. All five countries reported very few pulmonologists. Malawi, for example, had only one specialist respiratory clinician, who often did not receive referrals because of low awareness among healthcare workers.

### Poor reporting and reduced healthcare worker awareness

Low healthcare workers' diagnostic capacities implied that subsequent data inputs into the health information system may not provide reliable evidence of the CRD disease burden. Furthermore, the appropriate coding of diseases overall is undermined by shortage of health information officers and essential reporting tools.

**Chronic Respiratory Diseases in sub-Saharan Africa - Intersecting vicious cycles of neglect**

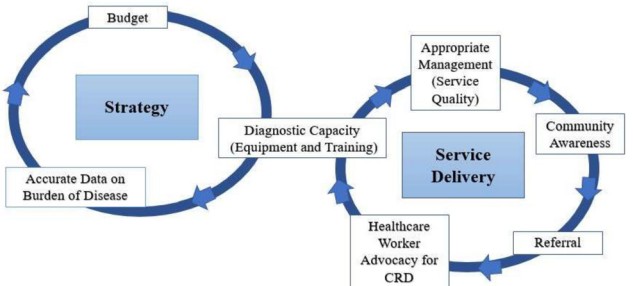

**Figure 1** A vicious cycle of neglect of CRD at the strategic policy level and service level. CRD, chronic respiratory disease.

…then you find most of our medical personnel record 'chronic cough' which is not a diagnosis … we are championing for this cadre of staff (health records information officers) because they are very critical otherwise, we would continue making wrong decisions based on poor data. (KII-non-state actor, Kenya)

This poor reporting was felt, in turn, to reduce awareness of CRD among healthcare workers of the need for clinical follow-up and of the overall scale of the problem.

## DISCUSSION

Based on data from five SSA countries (Kenya, Malawi, Sudan, Tanzania and Uganda), we identified how neglect of CRD at strategic and service levels forms two intersecting vicious cycles as illustrated in figure 1.

While specific weaknesses varied by context, common cycles were discernible. At strategic policy level, low diagnostic capacity, weak recording and reporting systems limited the availability of reliable data on the burden of lung diseases, negatively affecting inclusion in policy, and in turn budgetary allocations for diagnostic equipment, training and medicines. At the service level, lack of budgetary allocations for equipment and training of staff constrained diagnostic capacity, which, along with limited availability of appropriate medicines, reduced service delivery quality and collection of appropriate data within healthcare facilities. Providers felt frustrated at their lack of capacity to diagnose and treat these conditions; they were aware that that their own skills to support patients with CRD over time were limited.

To strengthen health systems in these countries and others in similar contexts, it will be important to break these vicious cycles,[25] although the most strategic entry points may vary by country. For example, variations in the development of policy strategies specific to lung health suggest policy change is more important in some contexts with NCD policy and TB control policy, both providing possible avenues of entry.[26] Improvements in diagnostic capacity, along with improving accuracy of reporting, may create impetus for policy commitments

and corresponding budgetary allocations, required to improve availability of medicines. Concurrent improvements in systems 'software' such as communication, advocacy and accountability are required to realise policy changes in practice.[27 28] For example, strategic communication of local data and accountability for responsiveness to this is needed, not only on burden of disease and service use but also on stockouts, broken equipment or vacant posts.[29] Ultimately, a team approach (including managers, clinical providers, health information officers and community actors) will be needed to ensure a coherent approach and feedback loops. The WHO, and civil society groups, such as professional thoracic societies and researchers, have an important role of developing a sustainable integrated response through their guidelines and practice in LMIC contexts. Effective communication of guidelines and protocols needs to be accompanied by operational plans to enact them. For example, it is now clear that treating patients with asthma with short-acting beta-agonists alone is associated with risk (exacerbation and thus likely deaths) even in mild asthma. This has led to a revision of the current Global Initiative for Asthma guidelines for asthma control advocate use of inhaled corticosteroid-formoterol for the management of mild asthma and yet many of the study countries restrict availability at the primary healthcare facilities most likely to see cases.[30]

It is likely to be too simplistic to assume that changes at one point in the cycle will lead to the desired changes in a linear way throughout the cycle, since challenges in a complex system are likely to be multicausal.[31] Evidence is only one, often small, element in policymaking.[32] In turn, written policy may not be translated into budgetary allocations or materialise as equipment and medicine availability due to intense budgetary competition; the role of donor priorities; 'leakage' of resources through the system; procurement and maintenance challenges, among others. Decentralisation offers opportunities for local (burden of disease) data and demands to influence budgetary allocations, and for enhancing primary healthcare services bringing CRD services closer to patients' residence. Decentralisation may however also mean greater political struggles over priorities[33 34] and increased fragmentation across the national health system. In many contexts, inadequate human resources and high staff attrition rates in rural areas provide a significant challenge to maintaining trained providers. Interventions in one part of the system may have unintended consequences in another[28]; where provider workloads are high, improved diagnostic capacity for certain diseases may deprioritise others, including by increasing service utilisation. Efforts will therefore be needed to track changes throughout the system, and beyond CRD.

Finally, management of CRD, like other chronic conditions, requires continuity of care, effective linkage between health facilities and community systems, and patient empowerment.[12 35] Significant communication is therefore required by providers, including community

health workers, to increase awareness of CRD and support patients with transition to self-management of their conditions.

Health systems in LMICs are currently designed to respond to acute, mostly communicable, illnesses usually through disease-specific vertical programmes. Despite some adaptation to treat people living with HIV, these systems often respond poorly to chronic conditions such as CRD. Key features of care models for other NCDs which could be adapted are having earmarked funding for organised and equipped healthcare teams, sustained supply of medicines, continuity of care, and strong linkages between healthcare facilities and community health systems. Integration of policy and services has benefits of a more holistic approach in addressing multimorbidity and can, for example, reduce poor prescribing practice.[36] Rather than adding a new vertical programme, or a new set of diseases to existing programmes, we advocate for a system-wide approach to a range of chronic diseases.

## Strengths and limitations

The main strengths and limitations both relate to the fact that the five studies were conducted independently, by different research teams with variations in the topic guides allowing for contextual insights. This allowed us to compare local strategies in dealing with CRD and share experiences and learn from them. The KII and IDI are appropriate in investigating and triangulating perspectives of study participants, enhancing rigour in the data collection. A larger number of district hospitals were sampled in Sudan and a focus on urban sites in Kenya, Malawi and Uganda with the inclusion of rural sites in Tanzania and Sudan. While these independently conducted studies established similar findings in diagnosis and management of CRD, suggesting important systemic challenges across the SSA countries, the findings should nevertheless be interpreted with caution, as environmental exposures and service delivery contexts may differ with the distribution of healthcare facilities and professionals relatively higher in urban settings. We did not include patients with CRD, community health workers and at-risk community members' perspectives in our study but focused on the overall issues that affect the planning and care of the patients. Due to constraints in data collection, saturation was not achieved in the Ugandan site.

## Recommendations and conclusions

The COVID-19 pandemic has brought into sharp focus the urgent need to reform and strengthen healthcare systems to effectively respond to people with chronic health conditions, including CRD, an important risk factor to COVID-19 severity. In the five SSA countries studied, interlinked gaps at the policy strategic level, and healthcare delivery levels, undermine appropriate provision of services for people with CRD. Lack of diagnostic capacity is a major link between the vicious cycles, influencing both the ability to manage cases within services and lack of accurate data on CRD to inform policy responses and

resource allocation. Improvements are required across all key elements of the service delivery systems, including pre-service and in-service training for diagnosis and management, guideline dissemination, diagnostic equipment, recording and reporting. Additionally, there is an urgent need to enhance reliable, affordable access to drugs, particularly the corticosteroid-formoterol inhaler in the early stages of asthma management in children, adolescents and adults.[37] Enhancing collection of population-level data to ascertain the true burden of disease may be an important entry point to drive policy change ensuring that people living with CRD are not 'left behind' in the development of universal healthcare.

**Author affiliations**
[1]International Public Health, Liverpool School of Tropical Medicine, Liverpool, UK
[2]Centre for Respiratory Diseases Research, Kenya Medical Research Institute (KEMRI), Nairobi, Kenya
[3]Lung Institute, Makerere University, Kampala, Uganda
[4]Health Systems and Policy Research Unit, REACH Trust Malawi, Lilongwe, Malawi
[5]National Institute of Medical Research, Mbeya, United Republic of Tanzania
[6]Lung Health Department, Epi-Lab, Khartoum, Sudan
[7]Medical Department, Kibong'oto Infectious Diseases Hospital/Kilimanjaro Christian Medical University, Kilimanjaro, United Republic of Tanzania
[8]Education Department, Liverpool School of Tropical Medicine, Liverpool, UK
[9]African Institute for Development Policy (AFIDEP), Nairobi, Kenya
[10]National Institute for Medical Research (NIMR), Mbeya Medical Research Centre, Mbeya, Tanzania, United Republic of
[11]AXESS Sexual Health, Liverpool University Hospitals NHS Foundation Trust, Liverpool, UK
[12]Tropical Infectious Diseases Unit, Liverpool University Hospitals NHS Foundation Trust, Liverpool, UK

**Collaborators** On behalf of the NIHR International Multidisciplinary Programme to Address Lung Health and TB in Africa (IMPALA) Consortium: Emmanuel Addo-Yobo, Brian Allwood, Hastings Banda, Imelda Bates, Amsalu Binegdie, Adegoke Falade, Jahangir Khan, Maia Lesosky, Bertrand Mbatchou, Kevin Mortimer, Beatrice Mutayoba, Louis Niessen, Jamie Rylance, S Bertel Squire, William Worodria, Heather Zar, Eliya Zulu, Lindsay Zurba.

**Contributors** SM—literature search, data collection, analysis, interpretation and writing. IA—literature search, data collection and analysis, interpretation and writing. NK—literature search, data collection and analysis, and writing. EHS—data collection and analysis, and writing. EA—data collection and analysis, and writing. RO—data collection and analysis, and writing. SGM—data collection and analysis, and writing. RT—study design, data analysis, interpretation and writing. UE—study design, interpretation and writing. MC—study design, interpretation and writing. KRC—data analysis, interpretation and writing. AE—interpretation and writing. HM—interpretation and writing. RO—interpretation and writing. NEN—interpretation and writing. AO—study design, interpretation and writing. MT (guarantor)—study design, data analysis, interpretation and writing.

**Funding** This research was funded by the NIHR IMPALA (grant reference 16/136/35) using UK aid from the UK Government to support global health research.

**Disclaimer** The views expressed in this publication are those of the author(s) and not necessarily those of the NIHR or the UK Department of Health and Social Care.

**Competing interests** None declared.

**Patient and public involvement** Patients and/or the public were not involved in the design, or conduct, or reporting, or dissemination plans of this research.

**Patient consent for publication** Not required.

**Ethics approval** This study involves human participants and the Liverpool School of Tropical Medicine Ethics Committee approved these studies separately (Kenya: protocol 18-054; Uganda: protocol 18-037; Malawi: protocol M1803; Tanzania and Sudan: protocol 18-043). Additionally, each approval was by in-country committees: Kenya Medical Research Institute (KEMRI/SERU/CRDR/037/3717); Malawi National

Health Science Research Committee (Protocol #18/04/2021); Sudan (Ref 44/T/Kh/1); Tanzania (Medical Research Coordinating Committee of the National Institute for Medical Research; NIMR/HQ/R.8a/V.IX/2922); and Uganda (TASO; IRB:TASOREC/030/18-UG-REC-009 and the Uganda National Council for Science and Technology HS232ES). All participants gave written, informed consent.

**Provenance and peer review** Not commissioned; externally peer reviewed.

**Data availability statement** Data are available upon reasonable request. No additional data available.

**ORCID iDs**
Stephen Mulupi http://orcid.org/0000-0001-8533-578X
Stellah G Mpagama http://orcid.org/0000-0002-0660-6930
Angela Obasi http://orcid.org/0000-0001-6801-8889

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
