## [Reviewer comments · BMJ Open]

ARTICLE DETAILS

TITLE (PROVISIONAL)	What are the barriers to the diagnosis and management of chronic respiratory disease in sub-Saharan Africa? a qualitative study with healthcare workers, national and regional policy stakeholders in five countries
AUTHORS	MULUPI, STEPHEN; Ayakaka, Irene; Tolhurst, Rachel; Kozak, Nicole; Shayo, Elizabeth; Abdalla, Elhafiz; Osman, Rashid; Egere, Uzochukwu; Mpagama, Stellah; Chinouya, Martha; Chikaphupha, Kingsley; Elsony, Asma; Meme, Helen; Oronje, Rose; Ntinginya, Nyanda; Obasi, Angela; Taegtmeier, Miriam

VERSION 1 – REVIEW

REVIEWER	Bell, Scott The Prince Charles Hospital
REVIEW RETURNED	20-Jul-2021

GENERAL COMMENTS	This is an interesting qualitative methodology based analysis of stakeholder reviews in five sub-Saharan African countries on management of chronic lung disease. The authors use key-informant interviews and in-depth interviews in 2018 and 2019 which included across the five countries – Kenya, Malawi, Sudan, Tanzania and Uganda, interviews of 60 policy stakeholders and 49 healthcare workers. The key findings were that the health systems were poorly equipped to respond specifically to burden of chronic lung disease which was under-diagnosed, under-reported and management underfunded. Key messages: 1. What is the role of NGO's, WHO researchers in developing a sustainable chronic lung disease response currently or potentially in the future.2. Asthma in many parts of the world is now managed in primary care. What are the implications of the findings for disease detection, education and management when access to limited experience and methods of diagnosis and management available close to patients' residence?3. What are the key features of current NCD models which could be adapted for chronic disease?4. What is the approach that the authors would suggest which would help mitigate the key findings of their study?5. In the post pandemic era, how should the health services respond from a policy and an access to therapeutics perspective?6. Under recommendations, Page 14, what did the authors feel are the roles of professional thoracic societies including the ATS, ERS and BTS in helping support universal health care for patients with chronic lung disease.
---

REVIEWER	Mehmood, Hana Maternal, Neonatal and Child health research network, Public Health
REVIEW RETURNED	09-Oct-2021

GENERAL COMMENTS	Kindly revisit the methodology section as it requires clarity for a number of elements. Attaching the documents with comments to be addressed
---

REVIEWER	Mroueh, Salman American University of Beirut, Pediatrics and Adolescent Medicine
REVIEW RETURNED	11-Oct-2021

GENERAL COMMENTS	The manuscript is well written. It attempts to answer an important question. And although pooling data from 5 countries may be looked at as a limitation, it is actually a strength, as it allows to compare and contrast local strategies in dealing with NCLD, and share experiences and learn from them. Question: Why did the authors cite the availability of the inhaled corticosteroid-formoterol inhaler as a priority in their conclusion, although its place in the management of asthma is still debatable, particularly in children?
---

REVIEWER	Ellington, Laura University of Washington, Pediatrics
REVIEW RETURNED	18-Oct-2021

GENERAL COMMENTS	Well written, important, and timely work to elaborate on challenges with NCD management in SSA. Minor points for clarification and fill in gaps below: Abstract - Please clearly report the study design/analysis Intro - well written, highlights pertinent background - Please clearly state the purpose of the study and specific objective(s) or questions. What is the reason for the presented study? What are the gaps to existing research? Methods - Per SRQR guidelines, please state qualitative approach - Please elaborate on the study team (see recommendations from SRQR guidelines), including a statement of reflexivity - Sampling: did you recruit until saturation achieved or other limitation? - Please provide interview guide as appendix/supplemental information - The intercountry analysis is unclear at what stages analysis took place. Were codes and topics identified within each country, then combined? Who participated in the coding meetings and determination of the final codebook? - It is unclear who received “common training in policy.” Participants, coders, or both? (paragraph on intercountry analysis) - Readers may not be closely familiar with the WHO health systems building blocks or how the 6 building blocks are used to frame the results. It would be helpful to list/summarize what these are since they are central to your results/organization.
--

	Results  - Consider at least one illustrative quote for every sub-heading. Some sub-heading have none currently. Discussion  - Clearly outline strengths - Consider moving figure to the results
--	---

VERSION 1 – AUTHOR RESPONSE

REVIEWER 1		
Comment #1	What is the role of NGO's, WHO researchers in developing a sustainable chronic lung disease response currently or potentially in the future	Thank you for this important point. We have addressed this in the second paragraph of the discussion where we have added” ‘The World Health Organization, and civil society groups, such as professional thoracic societies and researchers have an important role of developing a sustainable integrated response through their guidelines and practice in LMIC contexts.’
Comment #2	Asthma in many parts of the world is now managed in primary care. What are the implications of the findings for disease detection, education and management when access to limited experience and methods of diagnosis and management available close to patients’ residence?	This is noted and we agree primary care for CRD is important, although largely absent in our settings. We have expanded the text on decentralisation in the third paragraph of the discussion that begins: ‘It is likely to be too simplistic to assume...’ (3 rd paragraph discussion session)
Comment #3	What are the key features of current NCD models which could be adapted for chronic disease?	We believe the key features are: having earmarked funding for organized and equipped healthcare teams, sustained supply of medicines, continuity of care, and strong linkages between healthcare facilities and community health systems. We have addressed this through additional edits in the last paragraph of the discussion that begins ‘health systems in LMICs..’ (last paragraph before ‘strengths and limitation in discussion’).
Comment #4	What is the approach that the authors would suggest which would help mitigate the key findings of their study?	Thank you. We feel the key things are effective dissemination of existing guidelines with accompanying operational plans; improved communication between levels of the system and a system wide approach to chronic diseases. We have not further edited this text but bring the reviewer’s attention to the ends of paragraphs 2, 4 and 5 of the discussion respectively.
Comment #5	In the post pandemic era, how should the health services respond from a policy and an access to therapeutics perspective?	Thank you. We agree there is a need to strengthen healthcare services, but the policy response goes beyond access to therapeutics
Comment	Under recommendations, Page 14, what did the authors feel are the roles of	We have added some reflection on this in the second paragraph of the discussion

#6	professional thoracic societies including the ATS, ERS and BTS in helping support universal health care for patients with chronic lung disease.	along with the response to comment#1., on World Health Organization and professional societies
REVIEWER 2 Kindly revisit the methodology section as it requires clarity for a number of elements. Attaching the documents with comments to be addressed		
Comment #1	The abstract must have a line or two on the background and rational of the study	We have added an introductory sentence to the abstract to give this background.
Comment #2	Abstract objectives state policymakers but you also mentioned the healthcare workers in your title	Thank you. We have now clarified this.
Comment #3	The title mentions chronic lung diseases. Kindly be consistent with the terms. Either use chronic lung diseases or non-communicable lung diseases.	We have revised this throughout to read chronic respiratory disease (CRD)
Comment #4	It would good to define here what NCLD means and give some examples	Thank you. We have revised the first paragraph of the introduction to read: Globally the most prevalent CRDs are asthma and chronic obstructive pulmonary disease (COPD) but the definition includes other non-infectious lung and airway disease such as bronchiectasis and post-TB lung disease.
Comment #5	The methods need to clearly indicate how the sampling took place, profile of moderators, where were the interviews conducted, what was the mode of data collection (Face to face or telephonic), reflexivity needs to be addressed. In short, the authors clearly need to indicate information on the research team that primarily collected data, their credentials, occupation, gender, relationship with participants, methodological theory, participant selection, sampling, non-participation, setting of data collection, presence of non-participants, interview guide, repeat interviews, field notes, duration, and data saturation	Thank you for this. Since the studies were in 5 contexts and methods varied slightly between them we were not able to address all the detail requested in the available space. We have revised the 'methods section' to better clarify how the sampling took place, the profile of interviewers and their positionality). We clarified that the interviews took place face to face in workplaces or private venues and took about 40 minutes on average. The semi-structured interview guides are attached and will be made available.
Comment #6	It appears that you have used the terms 'IDIs' and 'KIIs' interchangeably. Although there are minor differences between the two, however, it is recommended to use KII as these were al KIIs.	We used the term in-depth interviews for interviews with healthcare workers and the term key informant interviews for interviews with policy stakeholders and have clarified this in the text first paragraph in 'data collection' section.
Comment #7	Which analysis framework was used and why?	We used a framework approach to our analysis and have referenced this in the section on inter-country analysis.
Comment #8	Need to mention the tools used in Nvivo	Thank you. This has been added in the same section.
Comment 10	It seems from this theme that the focus has been primarily on [lack of diagnostic	In all of our countries the health system prioritises ruling out TB before other lung

	equipment for JTB. It would be best if the authors could clarify if that was an intentional focus and why or was this coming out of the respondents. Either ways a justification on more focus on TB needs to be	diseases were investigated and our findings reflect this. Paragraph 1, " Lack of diagnostic equipment "
Comment 11	Limitations section: Why not include the strengths as well?	Thank you. We have edited this section to include strengths associated with including 5 countries
Comment 12	PPI declaration on patient involvement: Would be good to know how this was done. How many patients, their profiles, process of involvement, were all included in all stages or they were different in each stage etc.	Apologies. This was an error. Patients and the public were NOT involved in the design of the study. Appropriate changes made.
REVIEWER 3		
Comment #1	And although pooling data from 5 countries may be looked at as a limitation, it is actually a strength, as it allows to compare and contrast local strategies in dealing with NCLD, and share experiences and learn from them.	Thank you. We have reflected on this in our strengths and limitations section
Comment #2	Why did the authors cite the availability of the inhaled corticosteroid-formoterol inhaler as a priority in their conclusion, although its place in the management of asthma is still debatable, particularly in children?	The latest GINA recommendations are clear that step 1 asthma treatment for everyone aged 12 and above should be ICS-formoterol prn, and this is on the WHO list of essential medicines ICS-formoterol – we have edited this in the second paragraph of the discussion and added this reference (Reddel, Bacharier et al. 2021).
REVIEWER 4		
Comment #1	Abstract - Please clearly report the study design/analysis	we have added that this was a qualitative study and that we used a framework approach to the abstract.
Comment #2	Please clearly state the purpose of the study and specific objective(s) or questions. What is the reason for the presented study? What are the gaps to existing research?	Purpose of study outlined in the last paragraph of the introduction section. Gaps to existing research are highlighted in the second paragraph of the introduction section.
Comment #3	Methods  i. Per SRQR guidelines, please state qualitative approach ii. Please elaborate on the study team (see recommendations from SRQR guidelines), including a statement of reflexivity iii. Sampling: did you recruit until saturation achieved or other limitation? iv. Please provide interview guide as 	 i. The qualitative approach-constructivist philosophical paradigm (para 1) i. We have expanded details on who conducted the interviews in each context

	appendix/supplemental information v. The intercountry analysis is unclear at what stages analysis took place. Were codes and topics identified within each country, then combined? Who participated in the coding meetings and determination of the final codebook? vi. It is unclear who received “common training in policy.” Participants, coders, or both? (paragraph on intercountry analysis) vii. Readers may not be closely familiar with the WHO health systems building blocks or how the 6 building blocks are used to frame the results. It would be helpful to list/summarize what these are since they are central to your results/organization.	i. Cited in limitations ii. These are attached iii. Highlighted in the intercountry analysis i. The researchers (sentence 1, intercountry analysis) i. Included in the first paragraph of findings
	Results Consider at least one illustrative quote for every sub-heading. Some sub-headings have none currently.	Thanks- we have added an excerpt under “Lack of donor prioritisation decreases budgetary allocation”
Comment #5	Discussion i. Clearly outline strengths ii. Consider moving figure to the results	We have added some strengths to the section on limitations. Since the figure frames the summary of the findings in the first paragraph of the discussion we have retained it here and think it is the best location in this manuscript

Reference

Reddel, H. K., L. B. Bacharier, E. D. Bateman, C. E. Brightling, G. G. Brusselle, R. Buhl, A. A. Cruz, L. Duijts, J. M. Drazen and J. M. FitzGerald (2021). "Global Initiative for Asthma (GINA) Strategy 2021– Executive summary and rationale for key changes." The Journal of Allergy and Clinical Immunology: In Practice.

VERSION 2 – REVIEW

REVIEWER	Bell, Scott The Prince Charles Hospital
--

REVIEW RETURNED	07-Jan-2022
GENERAL COMMENTS	The authors have done an excellent job with the changes they have made to the MS in light of the reviews. No further concerns.
REVIEWER	Mroueh, Salman American University of Beirut, Pediatrics and Adolescent Medicine
REVIEW RETURNED	07-Jan-2022
GENERAL COMMENTS	The authors have appropriately addressed my concerns.
REVIEWER	Ellington, Laura University of Washington, Pediatrics
REVIEW RETURNED	24-Jan-2022
GENERAL COMMENTS	The authors' responses have adequately addressed these reviewer's concerns.